# Clinical Outcomes and Quantitative HBV Surface Antigen Levels in Diverse Chronic Hepatitis B Patients in Canada: A Retrospective Real-World Study of CHB in Canada (REVEAL-CANADA)

**DOI:** 10.3390/v14122668

**Published:** 2022-11-29

**Authors:** Carla S. Coffin, Sarah Haylock-Jacobs, Karen Doucette, Alnoor Ramji, Hin Hin Ko, David K. Wong, Magdy Elkhashab, Robert Bailey, Julia Uhanova, Gerald Minuk, Keith Tsoi, Alexander Wong, Mang M. Ma, Edward Tam, Mayur Brahmania, Carmine Nudo, Julie Zhu, Christopher F. Lowe, Carla Osiowy, B. Cord Lethebe, Stephen E. Congly, Eric K. H. Chan, Angelina Villasis-Keever, Urbano Sbarigia, Curtis L. Cooper, Scott Fung

**Affiliations:** 1Cumming School of Medicine, University of Calgary, Calgary, AB T2N 1N4, Canada; 2Department of Medicine, University of Alberta, Edmonton, AB T6G 2R3, Canada; 3Department of Medicine, University of British Columbia, Vancouver, BC V6T 1Z4, Canada; 4Department of Medicine, University of Toronto, Toronto, ON M5G 2CV, Canada; 5Toronto Liver Centre, Toronto, ON M6H 3M1, Canada; 6Bailey Health Clinic, Edmonton, AB T5H 4B9, Canada; 7Department of Internal Medicine, University of Manitoba, Winnipeg, MB R3T 2N2, Canada; 8Department of Medicine, McMaster University, Hamilton, ON L8S 4L8, Canada; 9Department of Medicine, University of Saskatchewan, Regina, SK S7N 5A2, Canada; 10Pacific Gastroenterology Associates, Vancouver, BC V6Z 2K5, Canada; 11Multi Organ Transplant Unit, Department of Medicine, Division of Gastroenterology, Western University, London, ON N6A 3K7, Canada; 12Cité-de-la-Santé de Laval, Laval, QC H7M 3L9, Canada; 13Faculty of Medicine, Dalhousie University, Halifax, NS B3H 4R2, Canada; 14Division of Medical Microbiology, Providence Health Care, Vancouver, BC V6Z 1Y6, Canada; 15National Microbiology Laboratory, Public Health Agency of Canada, Winnipeg, MB R3E 3R2, Canada; 16Janssen Global Services LC, Raritan, NJ 08869, USA; 17Janssen Pharmaceuticals NV, 2340 Beerse, Belgium; 18Department of Medicine, Division of infectious Diseases, University of Ottawa, Ottawa, ON K1N 6N5, Canada

**Keywords:** functional cure, HBsAg loss, quantitative HBsAg, Canada, multiethnic

## Abstract

Background: Hepatitis B surface antigen (HBsAg) loss is associated with improved clinical outcomes for individuals with chronic hepatitis B (CHB); however, the effects of varying HBsAg levels on clinical outcomes in diverse cohorts are understudied. Methods: In this cross-sectional, multicentre, retrospective study, the data on adult subjects enrolled in the Canadian HBV Network with CHB seen from 1 January 2012 to 30 January 2021 with the treatment and virologic data within 1 year of HBsAg testing were analyzed. Patients were tested for HBsAg using qualitative (for HBsAg-negative samples) and/or commercial quantitative assays. Fibrosis or hepatic necroinflammation was determined by the liver stiffness measurement (LSM). The baseline data were summarized using descriptive statistics and compared by using univariable/multivariable analyses. Results: This study included 844 CHB patients, with a median age of 49.6 years (IQR 40.1–60.5), and 37% were female. In total, 751 patients (78.6%) had known ethnicity data, and 76.7% self-reported as Asian, 11.4% as Black, 6.8% as White, and 4.8% as other. Among the 844 patients, 237 (28.0%) were HBsAg (−) (<LLOQ), 190 (22.5%) had qHBsAg 1–100, 91 (10.8%) had qHBsAg 100–500, 54 (6.4%) had qHBsAg 500–1000, and 272 (32.2%) had qHBsAg >1000 IU/mL. Overall, 80% (682) had known HBeAg status at the last follow-up, and the majority (87.0%) were HBeAg-negative. In addition, 54% (461/844) had prior antiviral therapy, 19.7% of which (16.3, 23.7, *n* = 91) were HBsAg (−). The treated patients had a lower risk of cirrhosis (16.46, 95% CI 1.89–143.39, *p* = 0.01) or HCC (8.23, 95% CI 1.01–67.39, *p* = 0.05) than the untreated patients. A lower proportion of the HBsAg-loss group had cirrhosis (5.7% vs. 10.9%, *p* = 0.021) and HCC (0.9% vs. 6.2%, *p* = 0.001). Conclusion: In this retrospective, ethnically diverse cohort study, CHB patients who received antiviral therapy and/or had HBsAg loss were less likely to develop cirrhosis and HCC, confirming the results of the studies in less diverse cohorts. No association was found between the qHBsAg level and fibrosis determined with LSM. Individuals who achieved HBsAg loss had low-level qHBsAg within 1 year of seroclearance.

## 1. Introduction

The hepatitis B virus (HBV) is a major global pathogen. Approximately, 292 million people worldwide have chronic hepatitis B (CHB) [1]. Adults with CHB are at an increased risk of developing cirrhosis, and/or hepatocellular carcinoma (HCC), which together accounted for 887,000 deaths among people living with HBV in 2015 [2].

Oral nucleos(t)ide analogs (NAs) can effectively reduce HBV viral load and the risk of HCC and cirrhosis [3]. However, NAs cannot induce a virologic cure (defined as the eradication of the intranuclear viral episome or covalently closed circular DNA, (cccDNA)) and rarely achieve a functional cure (defined as the loss of the hepatitis B surface antigen (HBsAg) sustained for ≥6 months off treatment, with or without antibody to HBsAg (anti-HBs) seroconversion) [4]. HBV DNA rebound may occur after NA cessation, and some patients remain at an increased risk of HCC [5,6].

HBsAg can originate from integrated viral genomes and cccDNA and can be found in vast excess in the serum of the infected individuals in the form of subviral particles (circulating HBsAg) or infectious virions. Low-level serum HBsAg may persist even in patients with undetectable viral load, and the integrated HBsAg DNA remains in the liver even in those patients who achieve HBsAg loss [7]. Integrated viral genomes are especially abundant in hepatocellular carcinoma (HCC)-derived liver tissue [8].

A recent systematic review and meta-analysis of 56 published studies (352,381 person-years of follow-up) found low rates of HBsAg seroclearance in untreated and treated patients (pooled annual rate of ~1%) [9]. Similarly, low rates of HBsAg loss were found in another systematic review of 34 published studies [10] (303,754 person-years of follow-up), indicating improved prognosis independent of HBV treatment history. Overall, substantial statistical heterogeneity was noted across various studies (I^2^ = 97.49%), which included both academic and community cohorts from Asia-Pacific (*n* = 23), Europe, the Middle East, as well as North (*n* = 2) and South America [10]. A large retrospective, single-centre Asian cohort study showed that patients with end-of-treatment qHBsAg < 100 IU/mL were more likely to achieve HBsAg loss after stopping NA therapy [11]. Comparatively, few studies have analyzed the clinical benefits of HBsAg loss and/or analyzed the quantitative HBsAg levels associated with clinical outcomes in diverse/multiethnic cohorts.

The Canadian HBV Network (www.canadianHBVnetwork.ca, accessed 26 November 2022) enables multisite research on hepatitis B in 8/10 Canadian provinces. This collaboration provides a source of data for a diverse population of CHB patients, including Canadians living with CHB who were born in one of 45 countries [12].

The objective of this cross-sectional cohort study was to conduct a retrospective analysis of the clinical and virologic data of those patients enrolled in the Canadian HBV Network who achieved HBsAg loss, on or off treatment, compared with HBsAg-positive patients. We present a descriptive comparison of patients with varying quantitative HBsAg levels based on a retrospective chart review.

## 2. Methods

### 2.1. Study Design

The current study is an opportunistic, cross-sectional, descriptive study of the available clinical data on patients living with chronic hepatitis B entered in the Canadian HBV Network Registry who achieved HBsAg loss and/or with quantitative HBsAg testing. The Canadian HBV Network (www.canadianHBVnetwork.ca, URL accessed on 26 November 2022) includes academic and community clinical care centres for patients with HBV across 8/10 Canadian provinces, including 14 clinics that participated in the current study. All the subjects included in this analysis were identified at the participating clinics. De-identified data were entered into the Canadian HBV Network REDCap^®^ database housed within the data coordinating centre at the University of Calgary, following the approved protocols by trained data entry coordinators. Inclusion Criteria: Adults >18 years of age with a history of CHB who were previously HBsAg-positive (>6 months’ duration) but had a laboratory-confirmed loss of HBsAg since January 2012, and/or remained HBsAg-positive (qualitative assay, Abbott Architect ^®^ or Roche Elecsys^®^ HBsAg II), with or without qHBsAg levels (Abbott Architect^®^, sensitivity of <=0.05 IU/mL) were included. A cross-sectional, retrospective analysis was conducted using the most recent patient record submitted to the database. Exclusion Criteria: Those with coinfection with hepatitis delta virus (positive to HDV antibody), hepatitis C virus (positive to HCV antibody), or HIV (positive to HIV antibody), or who underwent liver transplantation were excluded. In total, 6882 unique patients are enrolled in the CanHepB retrospective registry, 533 of which were excluded due to coinfection (i.e., if they tested positive for antibody to hepatitis delta, hepatitis C, or HIV) or underwent liver transplantation. The retrospective study cohort included 844 HBV-monoinfected patients who achieved HBsAg loss or known qHBsAg levels.

To determine long-term outcomes, patients will also be recruited prospectively at the participating clinics (including patients retrospectively identified and included in the current study who gave consent to be contacted). All the subjects will be followed for a minimum of 3 years, with an optional extension for a longer-term follow-up to analyze the impact of the HBsAg clearance on the natural history of HBV. This parallel prospective study is ongoing, and data from these analyses will be reported elsewhere.

### 2.2. Data Elements

For the retrospective analysis, the demographic, clinical, and virologic data were analyzed based on the most recent record updated in the database. The data captured in the electronic case report form included demographics, family history, medical history, risk factors for HBV, investigations, treatment (HBV and other), and hepatic complications. The criteria for the diagnosis of cirrhosis and HCC were based on the Canadian Association for the Study of the Liver (CASL)/Association of Medical Microbiologists and Infectious Disease (AMMI) Canada guidelines [13]. The diagnosis of NAFLD was based on expert clinical diagnosis, including the risk factors for metabolic-associated liver disease (obesity/hyperlipidemia/metabolic syndrome/type 2 diabetes), the absence of significant alcohol consumption, and/or an ultrasound showing confirmed steatosis. Recently, a revised nomenclature of metabolic-associated fatty liver disease (MAFLD) has been proposed to highlight the pathogenesis of metabolic-syndrome-associated conditions. If a controlled attenuation parameter (CAP) score was available, this provided additional Appendix A for the diagnosis of MAFLD hepatic steatosis (i.e., CAP > 248 decibels per meter (dB/m)). Mild/moderate steatosis (10–66%) and severe steatosis (>66%) were defined as CAP 248 to 279 dB/M and CAP > 280 dB/m, respectively [14].

### 2.3. Statistical Analysis

As this was a non-interventional, descriptive study, no formal sample size was calculated. It was estimated that retrospective, clinical data could be collected for approximately 200 patients with HBsAg loss compared with HBsAg-positive patients.

Data for the continuous variables were summarized as mean, 95% confidence interval (CI), and count (*n*), and comparisons were performed using a one-way ANOVA test, where three variables were analyzed, and a *t*-test, where two variables were analyzed. The categorical variables were summarized using the proportion and count, and comparisons were performed using chi-squared tests. These statistical analyses were conducted using GraphPad Prism version 9.0.0. Missing data were excluded from tables and figures. Logistic regression models were used to identify the predictors of HBsAg loss in all the cohorts and were reported as adjusted odds ratio (aOR) with 95% CIs and were analyzed using R version 4.0.3. These models were adjusted for demographic, clinical, and laboratory values. Specifically, in our univariate analysis, we evaluated demographic variables (age, sex, birthplace, and ethnicity); comorbidities (including non-alcoholic fatty liver disease (NAFLD)); HBV treatment; laboratory variables (hepatitis B e antigen (HBeAg), ALT, and HBV DNA). The patients were identified as having elevated enzymes if their alanine aminotransferase (ALT) values were >ULN (≥35 U/L for males and ≥25 U/L for females).

Between-group differences were evaluated with non-parametric tests, including the Wilcoxon signed-rank test and Mann–Whitney. A multivariable logistic regression analysis model (adjusted for sex, age, stage of fibrosis, and treatment) was used to evaluate the effect of confounding variables on the clinical, virologic, and immunologic outcomes. A type 1 error rate of 5% was used.

### 2.4. Ethics

The protocol was reviewed and approved by the University of Calgary Conjoint Research Ethics Board (#REB16-0041 and #REB19-2038). The patients provided written informed consent for their extracted de-identified data to be used or were included with a waiver of consent, according to the local Research Ethics Board (REB) approval. All the clinical and laboratory assessments directed by the treating physician were conducted according to the established protocols, as per the standard of care and expert guidelines. This study was conducted according to the Canadian and international standards of Good Clinical Practice.

## 3. Results

### 3.1. Patients

Data were collected from the most recent records available in 844 patients retrospectively identified who were either HBsAg-negative or HBsAg-positive, with or without quantitative HBsAg testing (Table 1). Previous studies have reported that low-level qHBsAg (i.e., <1000 IU/mL), and especially if <100 IU/mL, undetectable HBV DNA is associated with a reduced risk of HCC and disease progression [15,16]. Thus, the subjects in this study were grouped based on their qHBsAg levels, and the demographic and clinical data were compared between the groups. In total, 237 (28.0%) were HBsAg-negative (−) (<LLOQ), 190 (22.5%) had qHBsAg levels 1–100, 91 (10.8%) had qHBsAg 100–500, 54 (6.4%) had qHBsAg 500–1000, and 272 (32.2%) had qHBsAg levels > 1000 IU/mL. Ethnicity was available or self-reported in 88% of the patients (*n* = 751/844), 76.7% of which self-reported as Asian, 11.4% were Black, 6.8% were White, and 4.6% identified as other or unknown. There were no significant demographic differences between the groups.

### 3.2. Summary of Virological Outcomes

The virological data were analyzed based on the most recent patient record submitted to the database. HBeAg status was known in 80% (682/844) of the patients, *n* = 598 (87%) of which were HBeAg-negative at the last follow-up. In this real-world study, historical HBeAg tests were unknown; hence, the baseline hepatitis B disease activity (i.e., phase) could not be determined. However, in most HBeAg-negative individuals who start on antiviral therapy, repeat HBeAg and HBeAb testing is not performed, as treatment response can be assessed by HBV DNA and ALT levels. As expected, there were more HBeAg-negative patients in the HBsAg-loss group than in the groups who remained HBsAg-positive (*p* < 0.01). Similarly, more patients who achieved HBsAg loss had undetectable HBV DNA (72.5%) than patients with HBsAg < 100 IU/mL (49.3%) and HBsAg ≥ 100 IU/mL (21.1%, *p* < 0.01) (Table 2).

### 3.3. Summary of Antiviral Therapy and Outcomes

Based on the last record submitted to the database, 54.6% (*n* = 461) of all the patients included in this study had received treatment for CHB at any time. This included 91/237 (38.4%, 32.2, 44.9) of those patients who achieved HBsAg loss and who had received antiviral treatment (Table 3). The long-term data on treatment indication, the stage of the disease, response to treatment, and adherence were not available in this retrospective study. The majority (62.5%) of the treated patients received a tenofovir-based regimen (any regimen that contains tenofovir disoproxil fumarate or tenofovir alafenamide fumarate). Overall, 14% (66/461) of the treated patients received interferon, but no difference was noted between the cohorts (HBsAg loss or varying HBsAg levels) (Table 2 and Table 3). No patients received combination nucleos(t)ide analog and interferon therapy. Multivariate analysis revealed that HBsAg-positive patients who had received antiviral treatment for HBV at any point were less likely to develop cirrhosis (16.46, 95% CI 1.89–143.39, *p* = 0.01) or HCC (8.23, 95% CI 1.01–67.39, *p* = 0.05) than those patients who were never treated (Appendix A).

### 3.4. Comparison of Hepatic Outcomes between Groups

The patients who achieved HBsAg loss had lower mean ALT levels (25.6 U/L) than patients with qHBsAg < 100 IU/mL (30 U/L) and qHBsAg ≥100 IU/mL (42.8 U/L) (Table 2). Hepatic fibrosis and necroinflammation based on liver stiffness measurement were lower in the HBsAg-loss group than in those with HBsAg < 100 IU/mL (5.8 kPa, 95% CI, 5.2–6.3, vs. 8.4 kPa, 6.5–10.4), respectively, (*p* = 0.005), but no difference was observed from those with HBsAg ≥ 100 IU/mL (6.0 kPa, 5.5–6.5). The proportion of patients with cirrhosis (5.7% versus 10.9%, *p* = 0.021) and hepatocellular carcinoma (0.9% versus 6.2%, *p* = 0.001) was lower in those patients who achieved HBsAg loss than in patients with detectable HBsAg, respectively (Figure 1). Multivariate analysis showed that there was a strong association between the presence of cirrhosis and the development of hepatocellular carcinoma (odds ratio, 5.62; 95% CI, 2.04–15.48, *p* < 0.01). Interestingly, engaging in high-risk activities for acquiring blood-borne viral infection (i.e., recreational drug use, high-risk sexual contact, unsafe tattoos or piercings, etc.) (11.28; 1.57–80.86, *p* = 0.02) was also associated with developing HCC. The proportion of patients with non-alcoholic fatty liver disease (i.e., metabolic-associated fatty liver disease) was similar between the groups (HBsAg loss, 17.6%; HBsAg < 100 IU/mL, 26.5%; HBsAg ≥ 100 IU/mL, 19.4%, *p* = 0.048) (Table 1).

### 3.5. Other Factors Associated with HBsAg Loss

In this study, low qHBsAg levels, (measured within 12 months of HBsAg seroclearance) were associated with HBsAg loss (Figure 2). Individuals who reported participation in high-risk activities for blood-borne virus exposure (i.e., intravenous drug use), suggesting a horizontal mode of HBV transmission, were associated with higher rates of HBsAg loss. However, neither age nor self-reported White ethnicity or country of birth was correlated with HBsAg seroclearance. Osteoporosis, diagnosed based on the Canadian clinical practice guidelines [17], was less common in the HBsAg-loss group (0.9%) than in the HBsAg < 100 IU/mL group (4.4%), *p* = 0.025.

## 4. Discussion

This study describes the results of a retrospective, real-world analysis of the clinical outcomes in a diverse CHB patient population who achieved HBsAg loss (i.e., a functional cure) compared with individuals with varying HBsAg levels. The majority of patients who achieved HBsAg loss were untreated, HBeAg-negative, and had undetectable HBV DNA levels and normal liver enzymes (ALTs). In this study, individuals with HBsAg loss were also found to have low-level qHBsAg (<100 IU/mL), tested within 1 year of HBsAg seroclearance. Similarly, other single centre cohorts and real-world studies have reported higher rates of HBsAg loss in untreated CHB patients, especially with persistently (serial) low-level HBV DNA (<2000 IU/mL) and low qHBsAg levels [9,18,19,20,21]. In other studies, individuals on nucleos(t)ide analog therapy had a slow decline in HBsAg levels or only showed a marginal increase in the annual incidence of HBsAg loss [18,22]. Investigators have found that non-Asian ethnicity and/or White background is associated with treatment-induced or spontaneous HBsAg loss [23,24]. Our data are also consistent with the results from a recent, prospective study from the North American Hepatitis B Research Network (HBRN) that validated a simple, predictive score for HBsAg loss. In the HBRN cohort, HBsAg loss was associated with older age, non-Asian race, HBV phenotype (inactive carrier vs. others), HBV genotype A, lower HBV DNA levels, and lower titres and greater change in quantitative HBsAg levels [25]. Additionally, most of the subjects enrolled in the Canadian HBV Network and the HBRN cohort were born in HBV-endemic regions, including regions with greater socioeconomic disparities, and moved to higher-income regions. These sociodemographic factors can impact disease trajectory and long-term clinical outcomes including different lifestyles (i.e., adopting a more Western diet or more sedentary behaviours).

In the current study, hepatocellular carcinoma and cirrhosis were less common in patients with HBsAg loss than in patients with detectable HBsAg, which has also been reported by others [26,27]. The CHB patients in this study who achieve HBsAg loss were also less likely to have osteoporosis, independent of sex. Osteoporosis has been identified as a complication of CHB infection regardless of NA therapy such as tenofovir [27,28]. HBsAg loss was not associated with the presence of NAFLD or hepatic steatosis on ultrasound, unlike other reports [28,29]. However, our recent study within the Canadian HBV Network found that hepatic steatosis, which included a clinical assessment supported by a controlled attenuation parameter (CAP), was associated with undetectable HBV DNA; hence, further prospective follow-up may help clarify this relationship [14].

The current study found that severe liver fibrosis (usually determined by transient elastography or liver stiffness measurement) was less common in those patients who had HBsAg loss than in patients with qHBsAg < 100 IU/mL but not in those with qHBsAg ≥ 100 IU/mL. The lack of a correlation between qHBsAg levels and fibrosis stage has been reported previously [30,31]. Contrasting studies have found that both high HBsAg levels (>5000 IU/mL) [32] and very low-level HBsAg levels (<100 IU/mL) predict significant fibrosis [33]. We found that the patients in the HBsAg ≥ 100 IU/mL group tended to be younger than those in the HBsAg-loss group. Age has been identified as a major factor contributing to the development of viral hepatitis-induced fibrosis [34], although multivariate analysis did not show age to be a contributing factor in the diagnosis of cirrhosis in this study. A recent review reported that lower baseline qHBsAg (<100 IU/mL) can identify those individuals at a reduced risk of viral or biochemical flares, with serial annual declining titres and older age (>40 years) associated with HBsAg loss [35]. HBsAg can originate from multiple sources (i.e., integrated viral genomes, circulating subviral particles, and infectious virions). Integrated HBsAg can still be detected even in patients with functional cures and is highly present in malignant liver tissue [7]. Additional mechanistic studies are needed to fully understand the source of HBsAg. Long-term prospective follow-ups of patients with varying qHBsAg levels are needed to understand its influence on HBV-related liver disease and the natural history of CHB.

## 5. Limitations

These data only represent CHB patients who are engaged in clinical care in urban participating sites, which may impact the generalizability of these results. Thus, these findings may not be relevant to patients from non-specialist settings (especially rural communities). The HBV genotype and variants can impact HBsAg levels but were not available for most of the study participants. Our previous study also highlighted the genotype/ethnic diversity within the Canadian HBV Network [12]. HBV genotyping and sequence analysis is not performed as a routine standard of care by most clinics and is only available as a specialist referral diagnostic service by the National Microbiology Laboratory, Public Health Agency of Canada, Winnipeg, Canada. This is a retrospective, real-world study and hence limited by missing data and subject to selection bias. As a consequence, it was not possible to accurately classify CHB and follow the disease trajectory in the long term. There are few clinical cohort studies on hepatitis B on a national scale in North America. This study in a non-HBV-endemic country analyzes the clinical outcomes, together with quantitative HBsAg levels and liver stiffness measurement, in a diverse cohort. These data provide insight into patient clinical and treatment status at a single time-point and any treatment history, but details such as dosing changes or interruptions were not captured. Our ongoing prospective study of those patients who achieved HBsAg loss and/or with varying qHBsAg levels will provide important data to address these limitations.

## 6. Conclusions

In this real-world, multicentre, retrospective study of diverse patients with CHB, we found no association between qHBsAg levels and hepatic outcomes (i.e., liver stiffness measurement). Overall, the current work adds to the database of cross-sectional studies characterizing other HBV populations. We confirmed, in agreement with other studies performed in HBV-endemic countries and/or with less diverse cohorts, that antiviral treatment reduces the risk of severe liver disease. Additionally, persons who, likely after decades of chronic hepatitis B infection, achieved HBsAg loss showed a lower prevalence of hepatocellular carcinoma and cirrhosis than those with detectable HBsAg. Further prospective studies are being conducted by the REVEAL-CANADA investigators in the Canadian HBV Network, with a collection of biological samples to understand the natural history of CHB and the factors contributing to virological and clinical outcomes. These studies will advance the global efforts to achieve a hepatitis B cure.

## Figures and Tables

**Figure 1 viruses-14-02668-f001:**
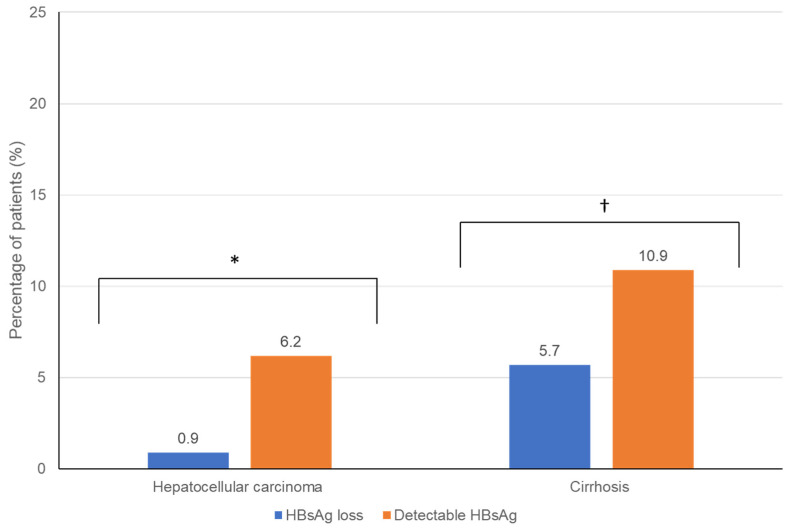
Percentage of CHB patients who achieved HBsAg loss vs. those remaining HBsAg-positive who developed the end-stage liver disease (i.e., HCC and cirrhosis). *****
*p* < 0.001. ^†^
*p* ≤ 0.001.

**Figure 2 viruses-14-02668-f002:**
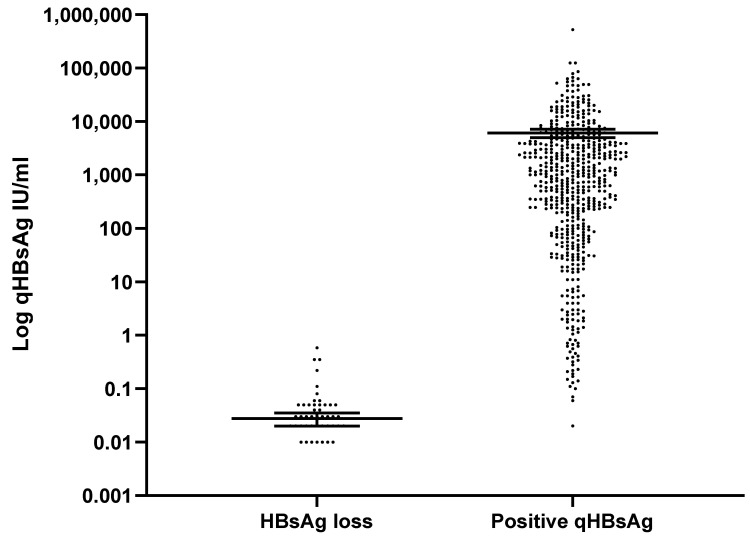
Quantitative HBsAg values in CHB patients who achieved HBsAg loss versus patients with positive HBsAg. HBsAg, hepatitis B surface antigen; IU, international units; qHBsAg, quantitative hepatitis B surface antigen.

**Table 1 viruses-14-02668-t001:** Summary of demographic and clinical data for CHB patients who cleared HBsAg vs. those remaining HBsAg-positive (e.g., quantitative (q)HBsAg 1–100, 100–500, 500–1000, and >1000 IU/mL).

Quantitative HBsAg Levels IU/mL (Sample Size) *n* = 844	HBsAg Neg (*n* = 237)	1–100 (*n* = 190)	100–500 (*n* = 91)	500–1000 (*n* = 54)	>1000 (*n* = 272)	*p*-Value HBsAg Neg vs. Pos.
Age, median (95% CI, n known)	53 (51.5, 54.6, 235/237)	55 (53.1, 56.8, 190)	55.4 (52.8, 58.1, 91)	52.5 (48.7, 56.2, 54)	44.1 (42.6, 45.6, 272)	<0.001 *
Female, % (95% CI, n)	37.6 (31.4, 44.1, 89/237)	34.7 (28.1, 42, 66/190)	34.1 (24.7, 44.8, 31/91)	33.3 (21.5, 47.6, 18/54)	41.2 (35.3, 47.3, 112/272)	0.701
Born in Canada, % (95% CI, n known)	3.4 (1.4, 7.6, 6/176)	2.1 (0.5, 6.4, 3/146)	1.5 (0.1, 9.4, 1/65)	6 (1.6, 17.5, 3/50)	5.4 (3, 9.5, 12/221)	0.442
Asian, % (95% CI, n known)	74.9 (68.2, 80.6, 149/199)	85.5 (78.9, 90.4, 136/159)	90.2 (81.2, 95.4, 74/82)	86.8 (74, 94.1, 46/53)	72 (66, 77.3, 185/257)	0.198
Black/African/ Caribbean (95% CI, n known)	13.1 (8.9, 18.7, 26/199)	7.5 (4.1, 13.1, 12/159)	7.3 (3, 15.8, 6/82)	9.4 (3.5, 21.4, 5/53)	20.2 (15.6, 25.8, 52/257)	0.071
White (95% CI, n known)	7 (4, 11.8, 14/199)	4.4 (1.9, 9.2, 7/159)	0 (0, 5.6, 0/82)	1.9 (0.1, 11.4, 1/53)	4.7 (2.5, 8.2, 12/257)	0.171
Other/ unknown ethnicity % (95% CI, n known)	5 (2.6, 9.3, 10/199)	2.5 (0.8, 6.7, 4/159)	2.4 (0.4, 9.4, 2/82)	1.9 (0.1, 11.4, 1/53)	3.1 (1.5, 6.3, 8/257)	0.536
Antiviral treatment for HBV at any time, % (95% CI, n known)	38.3 (32.2, 44.9, 91/237)	51.6 (44.3, 58.8, 98/190)	76.9 (66.7, 84.8, 70/91)	66.7 (52.4, 78.5, 36/54)	61 (54.9, 66.8, 166/272)	<0.001 *
Cirrhosis, % (95% CI, n known)	4.8 (2.4, 8.9, 10/209)	20.4 (14.6, 27.6, 33/162)	14.3 (8.1, 23.6, 13/91)	7.4 (2.4, 18.7, 4/54)	5.1 (3, 8.7, 14/272)	0.021 *
HCC, % (95% CI, n known)	1 (0.2, 3.8, 2/210)	9.2 (5.3, 15.2, 14/153)	7.7 (3.4, 15.7, 7/91)	9.3 (3.5, 21.1, 5/54)	2.9 (1.4, 5.9, 8/272)	0.001 *
NAFLD, % (95% CI, n known)	17.6 (12.9, 23.6, 37/210)	26.5 (20.1, 34.2, 43/162)	26.4 (17.9, 36.8, 24/91)	11.1 (4.6, 23.3, 6/54)	18.8 (14.4, 24, 51/272)	0.019 *

* For statistics comparing all groups, one-way ANOVA was used for continuous data, and chi-square tests were used for categorical data. ^†^ For statistics comparing HBsAg loss and low HBsAg groups, a *t*-test was used for continuous data, chi-square tests were used for categorical data, and *p* < 0.05 was considered significant. CI, confidence interval; HBsAg, hepatitis B surface antigen; HBV, hepatitis B virus; IU, international units; qHBsAg, quantitative hepatitis B surface antigen.

**Table 2 viruses-14-02668-t002:** Virological, biochemical, liver stiffness measurement (LSM), and antiviral treatment data of CHB patients who cleared HBsAg vs. those with low qHBsAg (<100 IU/mL) or with qHBsAg >100 IU/mL.

Clinical Data (% (95% CI, n Known))	HBsAg Negative (*n* = 237)	HBsAg 1–100 IU/mL (*n* = 190)	HBsAg ≥100 IU/mL (*n* = 417)	*p*-Value (HBsAg Loss vs. qHBsAg <100 IU/mL) ^†^
Laboratory				
qHBsAg level % (95% CI, n known)	0 (237)	20.1 (16.0, 24.1, 190)	8088 (5286.1–10891.8, 417)	N/A N/A
% HBeAg negative, (95% CI, n known)	98.9% (95.8, 99.8, 184/186)	94.1% (88.8, 97.1, 144/153)	78.9% (74.2, 83.1, 270/342)	0.004 *
HBV DNA, Log IU/mL (95% CI, n known)	0.21 (0.13–0.3, 182)	0.97 (0.76–1.17, 153)	2.15 (1.9–2.4, 378)	<0.001 *
ALT (U/L), (95% CI, n known)	25.9 (23.9–27.9, 224)	30.0 (26.1–33.9, 174)	42.8 (35.5–50.1, 373)	0.023 *
Liver Stiffness Measurement Using Transient Elastography (TE) (kPa)				
TE kpA (95% CI, n known)	5.7 (5.1-6.3, 109)	8.2 (6.5–9.9, 111)	6.0 (5.5–6.5, 302)	0.005 *
% >10.7 kpA (possible fibrosis stage 3)	6.7% (8/120)	17.5% (17/97)	4.3% (13/302)	0.013 *
Treatment 54.6% (*n* = 461/844)				
On treatment at any time, %	46.4% (110/237)	41.6% (79/190)	65.2% (272/417)	0.002 *
Tenofovir-based ^††^, %	27.8% (66/237)	25.3% (48/190)	41.7% (174/417)	0.027 *
Lamivudine, %	13.1% (31/237)	12.1% (23/190)	9.8% (41/417)	0.140
Entecavir, %	9.7% (23/237)	18.9% (36/190)	23.3% (97/417)	<0.001 *
Interferon, %	3.0% (7/237)	3.2% (6/190)	12.7% (53/417)	0.336
Months on treatment, mean (95% CI, n known)	84 (73.6–94.4, 87)	104.3 (90–118.5, 72)	87.6 (75.4–99.7, 260)	0.020 *

^†^ For statistics comparing HBsAg loss and low HBsAg groups, a *t*-test was used for continuous data, chi-square tests were used for categorical data, and * *p* < 0.05 was considered significant. For both continuous and categorical variables, missing data were excluded. ^††^ Tenofovir-based regimen refers to treatment regimen that contains tenofovir disoproxil fumarate (TDF) or tenofovir alafenamide (TAF). CI, confidence interval; HBsAg, hepatitis B surface antigen; IU, international units; qHBsAg, quantitative hepatitis B surface antigen.

**Table 3 viruses-14-02668-t003:** Comparison of demographic and clinical data in untreated vs. treated patients included in the study.

Variable	No Treatment (*n* = 383)	Treatment (*n* = 461)
Age	48.1 (46.8, 49.3) missing: 1/383	53.1 (51.9, 54.3) missing: 1/461
% Female	45.4 (40.4, 50.6, *n* = 174)	30.8 (26.7, 35.3, *n* = 142)
% Born in Canada	2.7 (1.3, 5.2, 9/333)	4.9 (2.9, 8, 16/325)
% Country of birth-Endemic *	54.4 (48.8, 59.8) 181/333	59.7 (54.1, 65) 194/325
% Asian	69 (63.9, 73.8) 243/352	87.2 (83.4, 90.2) 47/398
% Black	21.9 (17.7, 26.6) 77/352	6 (4, 9) 24/398
% White	4 (2.3, 6.7) 14/352	5 (3.2, 7.8) 20/398
% Other/not reported	5.1 (3.1, 8.1) 18/352	1.8 (0.8, 3.8) 7/398
% >2 Comorbidities **	9.4 (6.8, 12.9) 36/383	18 (14.7, 21.9) 83/461
% NAFLD	25.1 (20.7, 30) 87/347	16.7 (13.4, 20.6) 74/442
% Other cancer (not HCC)	2.1 (0.9, 4.4) 7/341	7.1 (4.9, 10) 31/439
% HBsAg negative	38.1 (33.3, 43.2) 146/383	19.7 (16.3, 23.7) 91/461
% HBsAg 1–100 IU/mL	24 (19.9, 28.7) 92/383	21.3 (17.7, 25.3) 98/461
% HBsAg >100 IU/mL	37.9 (33, 42.9) 145/383	59.0 (54.3, 63.5) 272/461

* Countries with ≥5% prevalence of HBV were considered endemic for HBV (Appendix A). ** The most common comorbidities reported were metabolic-associated fatty liver disease (MAFLD or NAFLD), hypertension, dyslipidemia, and diabetes (data not shown).

## Data Availability

Data supporting results are not publicly available.

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
