# Peer review of "Clinical Outcomes and Quantitative HBV Surface Antigen Levels in Diverse Chronic Hepatitis B Patients in Canada: A Retrospective Real-World Study of CHB in Canada (REVEAL-CANADA)"

_viruses, 2022, doi:10.3390/v14122668_

Round 1

Reviewer 1 Report

Carla S. Coffin and co-workers present a cross-sectional,  multi-center, retrospective study in a small cohort of 844 adult subjects enrolled in the Canadian HBV from January 1st 2012 to January 30th 2021 with treatment and virologic data within 1 year of HBsAg testing. Unfortunately their study design and conduct are inconsistent.  It is not reported how many subjects were excluded and how many were positive for hepatitis delta virus antibody and/or hepatitis C virus antibody and/or anti-HIV antibody. Since HBV genotypes influence significantly the serum HBsAg levels and the cohort is widely multi--ethnical  the study of the relations of qHBsAg with clinic-pathologic outcomes ha a major bias. Different therapy schedules can impact differently on HBsAg serum levels and not any information is given on the therapy schedules (how many were treated with NUCs vs IFNs vs combinations) and duration. Criteria used for the treatment decision making are not reported and furthermore both .baseline and follow-up data are very poor and mostly missing. Since the measure of liver stiffness provides a fibrosis inflammatory index the values of this variable should have been stratified for >/< normal serum transaminases for the statistical analysis of the relation of fibrosis with clinic-pathologic outcomes. The conclusion that individuals who achieved HBsAg loss had low-level qHBsAg within 1 year of  sero-clearance is not much of a new knowledge.  

Author Response

Reviewer 1

  • Carla S. Coffin and co-workers present a cross-sectional,  multi-center, retrospective study in a small cohort of 844 adult subjects enrolled in the Canadian HBV from January 1st 2012 to January 30th 2021 with treatment and virologic data within 1 year of HBsAg testing. 

Thank you for your comments.  We acknowledge that out study was limited by retrospective study design and missing information.  To address these limitations, study investigators within the Canadian HBV Network is currently conducting an ongoing multisite long-term prospective study of patients who achieved HBsAg loss or with low-level quantitative HBsAg levels.

  • their study design and conduct are inconsistent.  It is not reported how many subjects were excluded and how many were positive for hepatitis delta virus antibody and/or hepatitis C virus antibody and/or anti-HIV antibody.

Thank you. We have clarified the study design in methods and provide information on individuals excluded.  The current study is an opportunistic, cross-sectional descriptive study of

available clinical data on chronic hepatitis B included in the Canadian HBV Network Registry who achieved HBsAg loss and/or with quantitative HBsAg testing.  In total, 6882 unique patients are enrolled in the CanHepB retrospective registry, 533 patients were excluded due to co-infection (i.e., if they tested positive for antibody to hepatitis delta, hepatitis C and HIV) or underwent liver transplantation. The retrospective study cohort included 844 HBV monoinfected patients who achieved HBsAg loss or known qHBsAg level.

  • Since HBV genotypes influence significantly the serum HBsAg levels and the cohort is widely multi--ethnical  the study of the relations of qHBsAg with clinic-pathologic outcomes has a major bias.

Thank you.  We agree that HBV genotype can vary according to quantitative HBsAg levels as was found in our previous research studies conducted by our clinic and study investigators in the Canadian HBV Network (Conly et al., Liver International 2013, PMID  23763288; O’Neil et al., Annals of Hepatology 2018, PMID 31097238).  Our previous study also highlighted the genotype and ethnic diversity within the Canadian HBV Network (Coffin et al., CMAJ Open 2019, PMID 31641059).  However, HBV genotype is not done as routine standard of care by most clinics and is only available as a specialist referral diagnostic service by the National Microbiology Laboratory, Public Health Agency of Canada, Winnipeg, Canada). 

  • Different therapy schedules can impact differently on HBsAg serum levels and not any information is given on the therapy schedules (how many were treated with NUCs vs IFNs vs combinations) and duration. 

Thank you. We agree that HBsAg levels and decline can vary depending on prior therapy.  The collaborators involved in this study include major hepatology and infectious disease clinics across Canada. The co-authors agree that the vast majority (>95%) of people treated for hepatitis B in Canada receive nucleos(t)ide analogs and that interferon (Pegylated-Interferon) is rarely used due to both provider and patient preference.  There are provincial / regional variabilities in the availability of IFN which is not approved in several provinces (Congly and Brahmania, CMAJ Open 2019, PMID 30926602). No patients received combination therapy, and is not recommended as routine standard of care by Canadian HBV management guidelines (Coffin et al., Can. Liv J. 2018, PMID, 35992619).  Overall data from all patients in the Canadian HBV Registry indicate that <2% of those treated receive Interferon treatment. In the current study 66/461 (14%) of patients who received antiviral therapy, were treated with interferon (Table 3). However, this data does not reflect general clinical practice, and confounded by the fact that many of the same patients who receive interferon may also undergo HBsAg monitoring (quantitative HBsAg evaluation) to assess treatment response.

  • Criteria used for the treatment decision making are not reported and furthermore both

baseline and follow-up data are very poor and mostly missing. 

The authors appreciate the relevant and insightful comments raised by the reviewer.  Our ongoing prospective study will help address these limitations. The collaborators involved in the current study include experts in hepatology and infectious disease clinics in Canada. Treatment decisions are based on major guideline recommendations. We acknowledge that due to the cross-sectional retrospective study design, it was not possible to collect baseline and follow-up data.

  • Since the measure of liver stiffness provides a fibrosis inflammatory index the values of this variable should have been stratified for >/< normal serum transaminases for the statistical analysis of the relation of fibrosis with clinic-pathologic outcomes.

Thank you. We agree that liver stiffness measurement is affected by inflammation, as reflected by elevated transaminases (alanine aminotransferase, ALT).  In our study most patients in all groups had ALT <2x ULN based on ALT normal cut-off of 25 U/L for female and 35 U/L for male (Table 2). Our recent study with the Canadian HBV Registry cohort of patients who received antiviral therapy showed decline in liver stiffness measurement in patients on long-term antiviral therapy with normalization of ALT that can reflect improvement in liver inflammation as well as fibrosis regression (Ramji et al., World Journal of Gastroenterology 2022. PMID 36159017).  Very few patients underwent liver biopsy hence we are unable to correlate these findings to histopathological findings.

  • The conclusion that individuals who achieved HBsAg loss had low-level qHBsAg within 1 year of  sero-clearance is not much of a new knowledge.  

Thank you we appreciate the reviewers’ perspective but the authors respectfully disagree with this comment. The current study is of interest to the readership of Viruses and adds additional information in the context of a North American, resource-rich country, with a multi-ethnic and diverse population.  As noted by the other reviewers, the current study data “mirror reports of others and is a valuable contribution to the literature, adds to the database of other cross-sectional studies and lay the foundation for new therapies under aggressive development for HBV.” 

Reviewer 2 Report

In this cross-sectional retrospective study, Authors evaluated data on 844 adult subjects born in 45 different countries enrolled in the Canadian HBV Network.  They analyzed clinical and virologic data of patients who achieved HBsAg loss on or off treatment, compared to HBsAg-positive patients, and the correlation between quantitative HBsAg and clinical outcome. Results confirm data reported by other studies already published. However Authors recognize limitations related to missing clinical data and selected subjects bias. Furthermore they announced are planning  ongoing prospective study of this cohort patients to address these limitations.

Questions and comments:

1)      Are HBV genotypes known? It has been reported that HBV genotypes and mutations can influence the HBsAg quantitation

2)      The patient population analyzed migrated from countries with a low socio-economic standard to a country with a high socio-economic standard and different lifestyles. This can make a difference in terms of the outcomes of disease. You could argue this.

3)      How many patients with osteoporosis had received tenofovir therapy?

4)      NAFLD is a complex clinical condition that cannot be defined by simple CAP evaluation. New definition MAFLD has been recently proposed. I think it could be better simply report the CAP evaluation because complete data on NAFLD associated clinical conditions lack or  have not been reported

5)      Tables 1 and 2  are a bit busy and  confusing: try to simplify the reported data or to report in a different format

6)      Table 2: could you argue the different quantitative HBsAg values in lamivudine patients compared with tenofovir and entecavir?

7)      Table 3: comorbidities. Which are ?

Author Response

Response to Reviewer 2: 
In this cross-sectional retrospective study, Authors evaluated data on 844 adult subjects born in 45 different countries enrolled in the Canadian HBV Network.  They analyzed clinical and virologic data of patients who achieved HBsAg loss on or off treatment, compared to HBsAg-positive patients, and the correlation between quantitative HBsAg and clinical outcome. Results confirm data reported by other studies already published. However Authors recognize limitations related to missing clinical data and selected subjects bias. Furthermore they announced are planning  ongoing prospective study of this cohort patients to address these limitations.

Questions and comments:
1)      Are HBV genotypes known? It has been reported that HBV genotypes and mutations can influence the HBsAg quantitation 
Thank you, this is an important point that was also noted by Reviewer #1.   We agree that HBV genotype and HBV surface mutations (i.e., diagnostic escape variants) can vary according to quantitative HBsAg levels.  Previous study conducted by study investigators in the Canadian HBV Network evaluated HBV genotypes in association with quantitation HBsAg levels (Congly et al., Liver International 2013, PMID  23763288; O’Neil et al., Annals of Hepatology 2018, PMID 31097238).  The Canadian HBV Network has also conducted a retrospective study that highlighted the genotype and ethnic diversity within a subset of patients enrolled in the Canadian HBV Network registry (Coffin et al., CMAJ Open 2019, PMID 31641059).  However, HBV genotype and HBV sequencing for viral mutants (diagnostic escape variants, pre-core or basal core promoter variants, polymerase/reverse transcriptase or drug resistance) is not done as routine standard of care by most clinics. HBV sequence is only available as a specialist referral diagnostic service to the National Microbiology Laboratory, Public Health Agency of Canada, Winnipeg, Canada.  

2)      The patient population analyzed migrated from countries with a low socio-economic standard to a country with a high socio-economic standard and different lifestyles. This can make a difference in terms of the outcomes of disease. You could argue this. 
Thank you this is an important and relevant comment. We had added this point to the discussion.

3)      How many patients with osteoporosis had received tenofovir therapy?
As this was a cross-sectional retrospective study complete clinical data was not available on all co-morbidities such as the number of patients with osteoporosis who received tenofovir therapy. In the current study only 22 patients were recorded to have osteoporosis, and no association was found with tenofovir therapy.

4)      NAFLD is a complex clinical condition that cannot be defined by simple CAP evaluation. New definition MAFLD has been recently proposed. I think it could be better simply report the CAP evaluation because complete data on NAFLD associated clinical conditions lack or  have not been reported.
Thank you. This is a valid point which emphasizes the pathogenesis of metabolic syndrome in development of metabolic associated fatty liver disease. We have revised the methods to acknowledge that CAP alone is not sufficient to diagnose MAFLD and only used to provide supporting data for the diagnosis of MAFLD or NAFLD. We acknowledge the limitations of CAP, however it is widely used in our clinical practise to asses for hepatic steatosis. Our recent study conducted by the Canadian HBV Network identified association with CAP, obesity and low level viremia (Ko H-H et al., Gastro Hep Advances, 2022; 1:106), similar to other published studies.

5)      Tables 1 and 2  are a bit busy and  confusing: try to simplify the reported data or to report in a different format
Thank you.  We have removed some non-signficant data from Table 1 and a column from Table 2 to improve clarity

6) different quantitative HBsAg values in lamivudine patients compared with tenofovir and entecavir?
 Thank you. Unfortunately we do not have longitudinal robust data to evaluate qHBsAg levels associated with different antiviral therapy. 
7)      Table 3: comorbidities. Which are ?
The Canadian HBV Network registry includes data entry fields for comorbidity data including, diabetes, cardiovascular disease, hypertension, chronic kidney disease, osteoporosis, liver and other cancer, psychiatric, smoking, alcohol use extra-hepatic HBV manifestations (e.g. PAN, PCT, glomerulonephritis, lymphoma, vasculitis), hepatic comorbidities (e.g. NAFLD, PBC, PSC, autoimmune hepatitis, Wilson’s disease, Alpha-1 antitrypsin deficiency, hemochromatosis, alcohol related liver disease). However this real-world study and available dataset was limited due to retrospective study design. The most common co-morbidities reported was metabolic associated fatty liver disease (MAFLD or NAFLD), hypertension, dyslipidemia, and diabetes (data not shown). This statement is added as a footnote to Table 3.

Reviewer 3 Report

There is wide geographic, ethnic, virological, and pathological variation among HBV+ individuals.  HBsAg loss is considered the best clinical marker for spontaneous or therapy-induced functional cure of HBV infections, but the complexity of the clinical picture makes predicting who is most likely to clear HBV difficult unless the target population has been previously assessed epidemiologically.

 Here, Coffin et al take advantage of a large clinical network within Canada to do a detailed, real-world retrospective associational study of HBV patients in Canada.  They correlate a very wide range of relevant clinical variables with HBsAg clearance, and their conclusions largely mirror those obtained in other studies of other populations.  Consequently, novelty is modest. 

 Regardless of the limited novelty, I feel this work to be a valuable contribution to the literature because it both adds to the database of cross-sectional characterizations of HBV populations, and it permits extension of data from populations studied elsewhere into the Canadian population.  This may soon have practical implications because there are a large number of new therapies under aggressive development for HBV.  The first generations of these newer drugs are unlikely to be broadly curative, but the registration trials will provide some guidance regarding which patient subsets will be most likely to benefit from them.  Consequently, this study lays the foundation for enabling such personalized patient treatment decisions in the Canadian population.

Author Response

Reviewer #3.

There is wide geographic, ethnic, virological, and pathological variation among HBV+ individuals.  HBsAg loss is considered the best clinical marker for spontaneous or therapy-induced functional cure of HBV infections, but the complexity of the clinical picture makes predicting who is most likely to clear HBV difficult unless the target population has been previously assessed epidemiologically. Here, Coffin et al take advantage of a large clinical network within Canada to do a detailed, real-world retrospective associational study of HBV patients in Canada.  They correlate a very wide range of relevant clinical variables with HBsAg clearance, and their conclusions largely mirror those obtained in other studies of other populations.  Consequently, novelty is modest.  
Regardless of the limited novelty, I feel this work to be a valuable contribution to the literature because it both adds to the database of cross-sectional characterizations of HBV populations, and it permits extension of data from populations studied elsewhere into the Canadian population.  This may soon have practical implications because there are a large number of new therapies under aggressive development for HBV.  The first generations of these newer drugs are unlikely to be broadly curative, but the registration trials will provide some guidance regarding which patient subsets will be most likely to benefit from them.  Consequently, this study lays the foundation for enabling such personalized patient treatment decisions in the Canadian population.

Thank you. The authors greatly appreciate the positive comments of Reviewer #3 and have added some to our concluding statement.

Round 2

Reviewer 1 Report

Authors acknowledge in their manuscript the major limitations of their retrospective multi center and multi ethnical cohort study and actually no other amelioration can reasonably be asked.